# The effects of compression load to the trunk on lipid metabolism in an inactive phase

**Kousuke Shimada**[1]*, **Masakatsu Nohara**[1], **Fumika Shinozaki**[1], **Midori Tatsuda**[2], **Takayuki Watanabe**[2], **Asuka Kamei**[1]*, **Keiko Abe**[1,3]*

1 Group for Food Functionality Assessment, Kanagawa Institute of Industrial Science and Technology, Kawasaki, Kanagawa, Japan, 2 DIANA Co.,Ltd., Shibuya-ku, Tokyo, Japan, 3 Department of Applied Biological Chemistry, Graduate School of Agricultural and Life Sciences, The University of Tokyo, Bunkyo-ku, Tokyo, Japan

* kamei@kistec.jp (AK); kousuke.s0223@gmail.com (KS); aka7308@mail.ecc.u-tokyo.ac.jp (KA)

**Data Availability Statement:** All relevant data are within the manuscript and its Supporting information files.

## Abstract

The effects of compression load to a specific body part, e.g. leg, arm, or trunk, evoke many functions and are applied in various fields including clinical medicine, sports, and general health care. Nevertheless, little is known about the functional mechanism of compression load, especially regarding its effects on metabolic function. We investigated the effects of compression load to the trunk on the metabolism. We designed adjustable compression clothes for mice and attached them to ten-week-old C57BL/6N male mice in a controlled environment. The mice were divided into compression and no-compression groups, the latter only wearing the clothes without added compression. The evoked metabolic changes were evaluated using indirect calorimetry and transcriptomics with liver tissue to investigate the mechanism of the metabolic changes induced by the compression load. The results indicated decreases in body weight gain, food intake, and respiratory exchange ratio in the compression group compared to the no-compression group, but these effects were limited in the "light period" which was an inactive phase for mice. As a result of the transcriptome analysis after eight hours of compression load to the trunk, several DEGs, e.g., *Cpt1A*, *Hmgcr*, were classified into functional categories relating to carbohydrate metabolism, lipid metabolism, or immune response. Lipid metabolism impacts included suppression of fatty acid synthesis and activation of lipolysis and cholesterol synthesis in the compression group. Taken together, our results showed that activation of lipid metabolism processes in an inactive phase was induced by the compression load to the trunk.

## 1. Introduction

Several studies reported various functional effects of compression loads on a body, body parts, or cells. For instance, the cells which constitute bones, including osteoblasts and osteoclasts have a mechanosensing network [1, 2] and additional compression load to osteoblasts induce RANKL expression change and adjusts the bone morphological process [3–5]. Compression loads are used for several functional applications in various fields including clinical medicine,

**Funding:** This study was funded by DIANA Co.,Ltd. and this study was also supported by a research grant from JSPS KAKENHI Grant Numbers 15H05346 (to A. K.), 16K12734 (to K. S.), 18K05514 (to A. K.), 19K14047 (to K. S.).

**Competing interests:** This study was funded by DIANA Co.,Ltd. T.W and M.T are employees of DIANA Co.,Ltd. DIANA Co.,Ltd. also provided support in the clothes for this study. There are no products in development or marketed products associated with this research to declare. The sponsor had no control over the interpretation, writing, or publication of this study, and does not alter our adherence to PLOS ONE policies on sharing data and materials.

**Abbreviations:** DEGs, Differentially expressed genes; FDR, false discovery rate; NCBI, National Center of Biotechnology Information; RIN, RNA integrity number.

sports training, performance, and recovery, as well as general health care. Clinically, the compression load is used as one of the therapies for treating lower extremity varicose veins [6] and preventing lymphedema [7]. In sports, a number of recent studies have reported that compression load functions in reducing fatigue accumulated by exercise [8, 9]. The effectiveness of compression load in treating lower extremity varicose veins or reducing fatigue is due to how compression influences the vascular system, e.g., excitometabolism of waste products or vasodilator action [10, 11]. Moreover, a recent study has reported that compression garments influence cerebral blood flow circulation and cognitive function [12]. These papers suggest that a compression load to a specific site, e.g., leg, arm, or trunk, may influence not only those specific sites but also the whole body, because blood circulates through a whole body. In particular, metabolic functions lead to important mechanism for homeostatic maintenance in many organs, and these processes are related to the vascular system which is present in the whole body. Thus, we hypothesized that compression load may influence the metabolic function of the whole body. At present, little is known about the functional mechanisms of compression load, especially the effects on metabolic function. Therefore, in this study, we made adjustable compression clothes for mice; we aimed at evaluating the effects of the compression load on their trunks (from the clothes) on their metabolic function using indirect calorimetry and liver transcriptome analysis.

## 2. Materials and methods

### 2.1. Animals

Ten-week-old male C57BL/6N mice were purchased from Charles River (Hino, Japan). Mice were housed at a temperature of 22 ± 2 ˚C and a humidity of 50 ± 5% with a 12 hr light/dark cycle (lights on 8 a.m.–8 p.m). They had a free access to food and water, were provided with a chow diet (MF; Oriental Yeast, Tokyo, Japan) throughout the experimental period. The mice were acclimated in advance to the laboratory environment cages (24 cm length × 17.2 cm width × 12.9 cm depth, 413 cm$^2$ floor area) for five days. Body weight gains and food intakes were regularly measured every other day during the experimental period. The animal experimental committee of the Innovation Center of Nanomedicine (Kawasaki, Kanagawa, Japan) approved all the animal experimental protocols (permit no.: A18-001-2).

### 2.2. Attaching adjustable compression clothes to a mouse

Adjustable compression clothes for mice used in this study were made with an aid of DIANA Co.,Ltd. The established materials used for adjustable compression clothes are comprised of textile cloth of 140 denier filament from the chest to the waist; compression load can be adjusted according to the degree of textile overlap. Mice were anesthetized using 2.5% isoflurane at a flow rate of 1 L/min and then attached in the adjustable compression clothes. To minimize the effects of the anesthesia, the anesthesia periods were set at 3 min or shorter for each application. When we applied changes to the points of compression, the mice were similarly anesthetized. The degree of compression load to the trunk was determined by 10% reduction of waist circumference. The compression load value was measured using the pressure measuring system (AMI3037-SB; AMI-Techno, Tokyo, Japan) for the compression group (n = 20) and the no-compression group (n = 20) at the time of compression condition change. These clothes loosened over time because of losing body weight by compression, and had to be re-tightened under anesthesia to maintain the pressure. To avoid re-anesthesia repeatedly, since anesthesia has no small effect on metabolism, the experiments were conducted within a short time to maintain the clothes pressure.

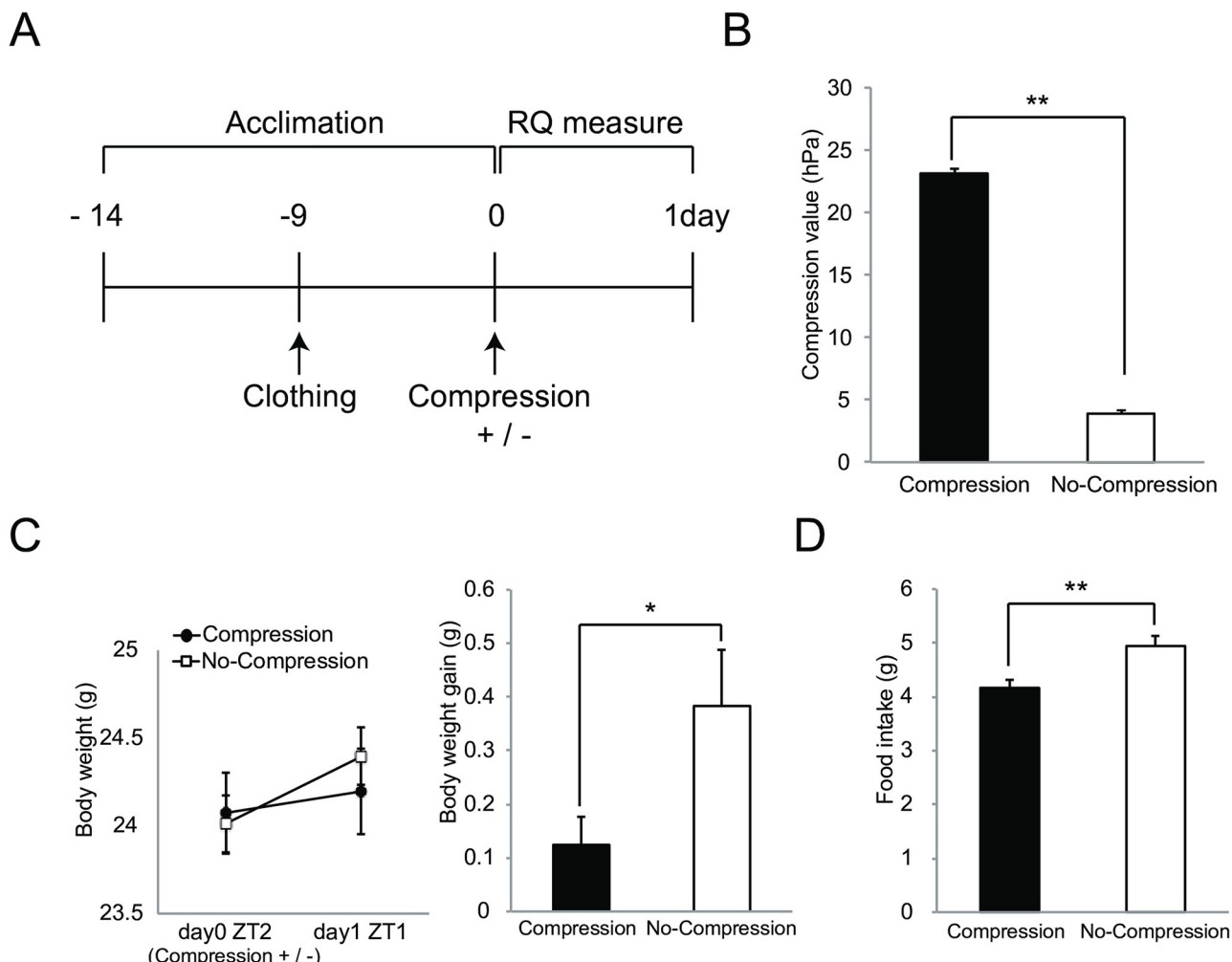

**Fig 1. Effect of clothes compression to the trunk.** A: Experimental protocol to investigate an effect on clothes compression load. B: Measure of compression value on left side of the abdomen by clothes compression load. C: Measurement of body weight change by clothes compression load to the trunk. (left panel showing body weight change and right panel is body weight gain). D: Measurement of food intake by clothes compression load. Each value is a mean with SEM. Significant difference detected by unpaired t-test. (*: $p < 0.05$, **: $p < 0.01$). B: n = 20 (no-compression group), n = 20 (compression group), C and D: n = 12 (no-compression group), n = 14 (compression group), respectively.

## 2.3. Measurement of indirect calorimetry

Mice were transferred to metabolic chambers with a size of 14 cm length × 14 cm width × 14 cm depth, 196 cm$^2$ floor area after four days of wearing the adjustable compression clothes. After five days of acclimation to the metabolic chambers, as depicted in Fig 1A at experimental day 0, respiratory gases, $O_2$ and $CO_2$, exhausted from each chamber were measured with an indirect calorimetric system (ARCO-2000; ARCO SYSTEM INC., Japan). The respiratory exchange ratio and energy expenditure were calculated based on $O_2$ consumption and $CO_2$ production. These gas productions from each chamber were analyzed every 3 min, and the obtained data was calculated for every 30 min average. In the respiratory gases analysed for 24 hours, the compression group (n = 14) and the no-compression group (n = 12) were compared, and the analysis for six hours, n = 7 in the compression-loaded and the no-compression group, respectively, were compared.

## 2.4. Sample preparation

Mice were anesthetized using 2.5% isoflurane at a flow rate of 1 L/min, and after cervical dislocation, the liver samples were collected. These liver sections were immediately frozen after excision using liquid nitrogen. All collected liver samples were maintained at −80˚C until use.

## 2.5. DNA microarray assay

The samples used for DNA microarray were n = 7 in the compression-loaded and the no-compression group, respectively. Total RNA was isolated from each hepatic sample with TRI Reagent(R) (Molecular Research Center, Inc, Cincinnati, OH, USA) and purified using an RNeasy mini kit (QIAGEN, Tokyo, Japan). The RNA integrity number (RIN) was estimated as an index of the quality of the total RNA using an Agilent 2100 Bioanalyzer (Agilent Technologies Japan, Tokyo, Japan); the RIN of our total RNA isolated from each liver was higher than 8.5. Then, cRNA was synthesized from 200 ng of purified total RNA, and sense-strand cDNA was obtained using a GeneChip™ WT PLUS Reagent Kit (Thermo Fisher Scientific Inc, Waltham, MA, USA). The sense-strand cDNA was fragmented, biotinylated and hybridized to a Clariom™ S array, mouse (Thermo Fisher Scientific Inc) at 45˚C for sixteen hours. The arrays were washed and stained with phycoerythrin with the GeneChip® Fluidics Station 450 (Thermo Fisher Scientific Inc). The arrays were submitted to scanning on an Affymetrix Gene-Chip® Scanner 3000 7G (Thermo Fisher Scientific Inc). The Affymetrix® GeneChip® Command Console® Software (Thermo Fisher Scientific Inc) was used to make CEL files. All DNA microarray data (.CEL files) presented in this publication have been deposited in the National Center of Biotechnology Information (NCBI) Gene Expression Omnibus (http://www.ncbi.nml.nih.gov/geo/). The data are accessible through accession number GSE150978.

## 2.6. DNA microarray data analysis

The CEL files were quantified by the qFarms using the statistical language R (3.4.4) (http://www.r-project) and Bioconductor (3.5) (http://www.bioconductor.org/). Hierarchical clustering was performed using the pvclust function in R. The rank products 2 (RP2) method was used to identify differentially expressed gene probe sets of the qFarms-quantified data. The probe sets with a false discovery rate (FDR) < 0.01 were considered to be differentially expressed genes (DEGs) between the compression and no-compression groups.

## 2.7. Ingenuity pathway analysis

Liver DEGs affected by the compression load to the trunk were submitted to QIAGEN's Ingenuity Pathway Analysis (IPA®, QIAGEN Redwood City, www.qiagen.com/ingenuity) and were functionally classified by disease and biofunctional analysis. The DEGs were coded as +1 (compression group > no-compression group) or −1 (compression group < no- compression group). Absolute Z-scores ≥ 2 were regarded as significantly enriched functions.

## 2.8. Statistical analysis

Each of data is presented as mean ± SEM. For physical and biochemical analysis, the data were compared by unpaired the Student's t-test at $p < 0.05$.

## 3. Results

We made adjustable compression clothes for mice that they were able to wear without inhibiting normal behavior, e.g., rearing, grooming, and hanging behavior. Fig 1A shows an experimental protocol for investigating an effect on metabolic function by compression load to the

trunk. Ten-week-old C57BL/6N male mice that were acclimated to the rearing environment and clothes were divided into compression-load and no-compression groups, the latter is a group of wearing the clothes without added compression. In this study, the degree of compression to the trunk was adjusted to the point that waist circumference was reduced by 10%. To measure the degree of compression value, a pressure measurement probe was set on the left side of the abdomen before compression load, and the compression value was measured after compression load (waist circumference 10% reduction). As a result, the values in the range of 20 to 25 hPa were confirmed on the left side of the abdomen by compression load to 10% waist circumference reduction (Fig 1B). To evaluate the influence on body weight change by compression load from wearing adjustable compression clothes, we confirmed body weight changes one day after the compression load. The effect of body weight change was not found in the average weight in each group, but body weight gain in the compression-load group was significantly inhibited in comparison with no-compression group (Fig 1C). Moreover, the amount of food intake during one day in the compression group was significantly lower than that in the no-compression group (Fig 1D). Next, to evaluate effects of the compression load on metabolism function, we analyzed respiratory gases ($O_2$ and $CO_2$) exhausted after compression load for one day using an indirect calorimetry system. The left panel of Fig 2A shows the mean plot of respiratory exchange ratio per one hour after changing a compression load

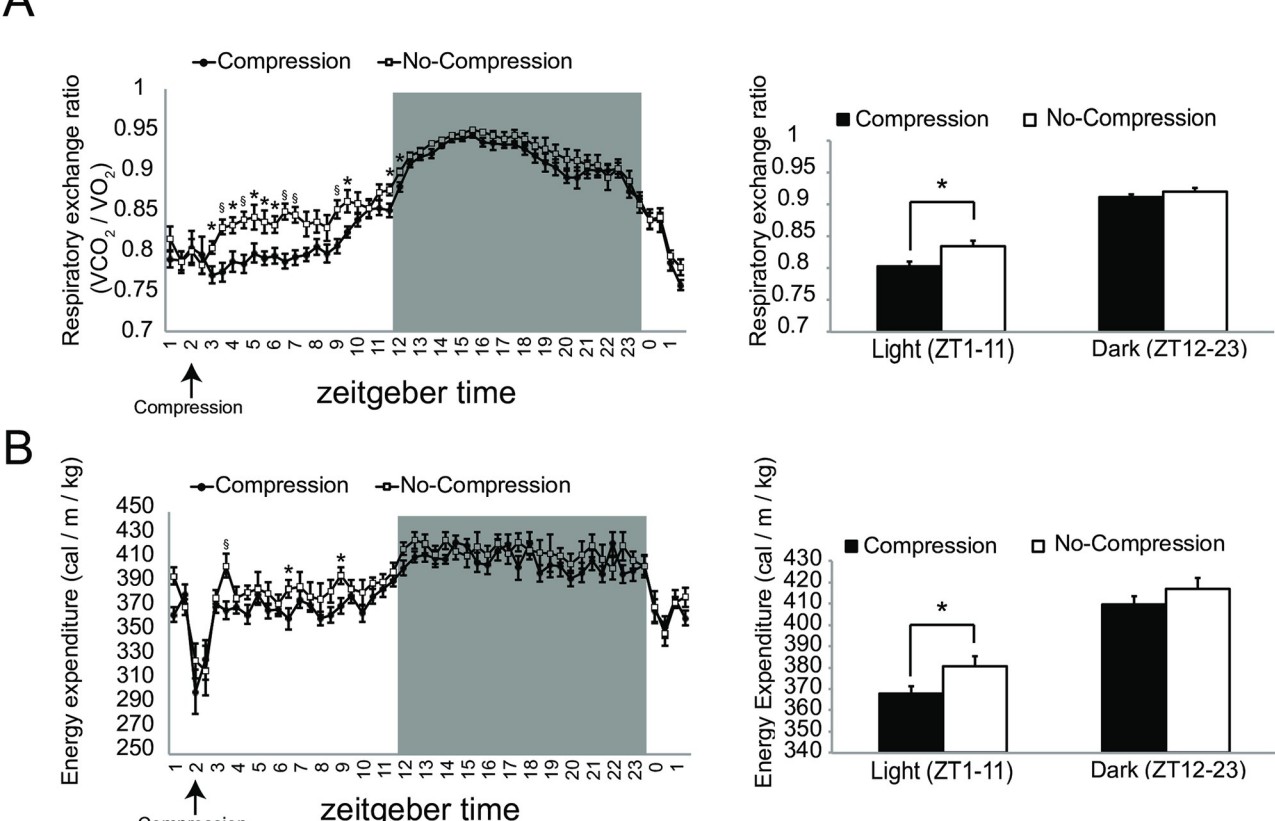

**Fig 2. Time course of change in respiratory quotient ratio (RQ) and energy expenditure using indirect calorimetry.** A: Left picture is a temporal change of respiratory exchange ratio (RER) after clothes compression load. Right picture is a mean of RER on light period (ZT1 to ZT11) and dark period (ZT12 to ZT23). B: Left picture is a temporal change of energy expenditure after clothes compression load. Right picture is mean of energy expenditure on light period (ZT1 to ZT11) and dark period (ZT12 to ZT23). Values are mean with SEM. Significant difference detected by unpaired t-test (*: $p < 0.05$ §: $p < 0.01$). n = 12 (no-compression group), n = 14 (compression group).

condition. Several time points of the respiratory exchange ratio of the compression group in the light period (ZT2 to ZT11) showed lower values than that for the no-compression group; average values in the light period were also lowering the compression group (Fig 2A right). On the other hand, differences in the effect on respiratory exchange ratio in the dark period (ZT12 to ZT23) were not found between the groups (Fig 2A). Next, we evaluated the effects of compression load on energy expenditure. Several time points of energy expenditure for the compression group in the light period (ZT2 to ZT11) and their average for the light period were lower than for the no-compression group (Fig 2B). Energy expenditure in the dark period (ZT12 to ZT23) between groups was not influenced, similar to the respiratory exchange ratio. According to respiratory gases analysis, we confirmed that the effect of the compression load to the trunk was caused in the light period (ZT2 to ZT11). Next, we focused on light period that the effect on compression load was expected, and investigated an effect on metabolic function, e.g., body weight change, amount of food intake, respiratory gases analysis, and transcriptome in the liver as the main metabolic organ, after short-term compression load. The acclimation period before compression load is depicted in Fig 1A. The respiratory gases analyses were performed during compression load at six hours; liver tissue was sampled eight hours after compression load, and then hepatic gene expression was analyzed (Fig 3A). Respiratory gases analysis showed that the respiratory exchange ratio of the compression group was lower than that of the no-compression group, as in the previous experiment (at one day) (Fig 3B). Body-weight loss in the compression group was significantly increased in comparison with the no-compression group after 8 hr of compression load to the trunk (Fig 3C). Also, food intake in the compression group was significantly lower in comparison with the no-compression group (Fig 3D). On the other hand, compression load for eight hours did not influence tissue weight (e.g. liver and epididymal adipose tissue) (S1 Fig). Next, we investigated an effect to gene expression in liver which was burdened of compression load. Collected liver samples were analyzed using DNA microarray, and differentially expressed genes (DEGs) between the two groups were predicted using the rank products method (FDR < 0.01). As a result, 604 DEGs (compression group > no-compression group: 334 DEGs, compression group < no-compression group: 270 DEGs) were statistically extracted. To elucidate a function regulated by gene expression in the liver caused by compression load to the trunk, we investigated functional enrichment analysis of DEGs using the Ingenuity Pathways Analysis (IPA, Ingenuity Systems). As a result of diseases and biological functions analysis of IPA, most of the enriched functions of DEGs in the liver were mainly classified in functional categories relating to carbohydrate, lipid metabolism, or immune response (Fig 4A). Functional categories relating to carbohydrate or lipid metabolism functions were predicted as active/increase direction in the compression group. On the other hand, functional categories relating to immune response functions were predicted to inhibit/decrease direction in the compression group. Several DEGs relating to cholesterol metabolism adjustment and beta-oxidation processes were included in categories relating to lipid metabolism. Further, in functional categories relating to lipid metabolism, DEGs relating to bile acid synthesis of secretion processes were also observed. In lipolysis processes, the gene expression of lipolysis enzymes Lipase E (Lipe) and Lipase C (Lipc) were up-regulated in the compression group. In cholesterol metabolism, the gene expression of ATP-binding cassette transporters ATP-Binding Cassette Subfamily A Member 1 (Abca1) and ATP-Binding Cassette Subfamily B Member 11 (Abcb11) and Acetyl-CoA Acetyltransferase 2 (Acat2) were down-regulated in the compression group. In fatty acid metabolism, the gene expression of beta-oxidation enzymes, Acyl-CoA Synthetase Long-Chain Family Member 1 (Acsl1), Carnitine Palmitoyltransferase 1A (Cpt1a), and 3-Hydroxy-3-Methylglutaryl-CoA Synthase 2 (Hmgcs2) were up-regulated, and bile acid synthesis enzymes, Cytochrome P450 Family 7 Subfamily A Member 1 (Cyp7a1), Cytochrome P450

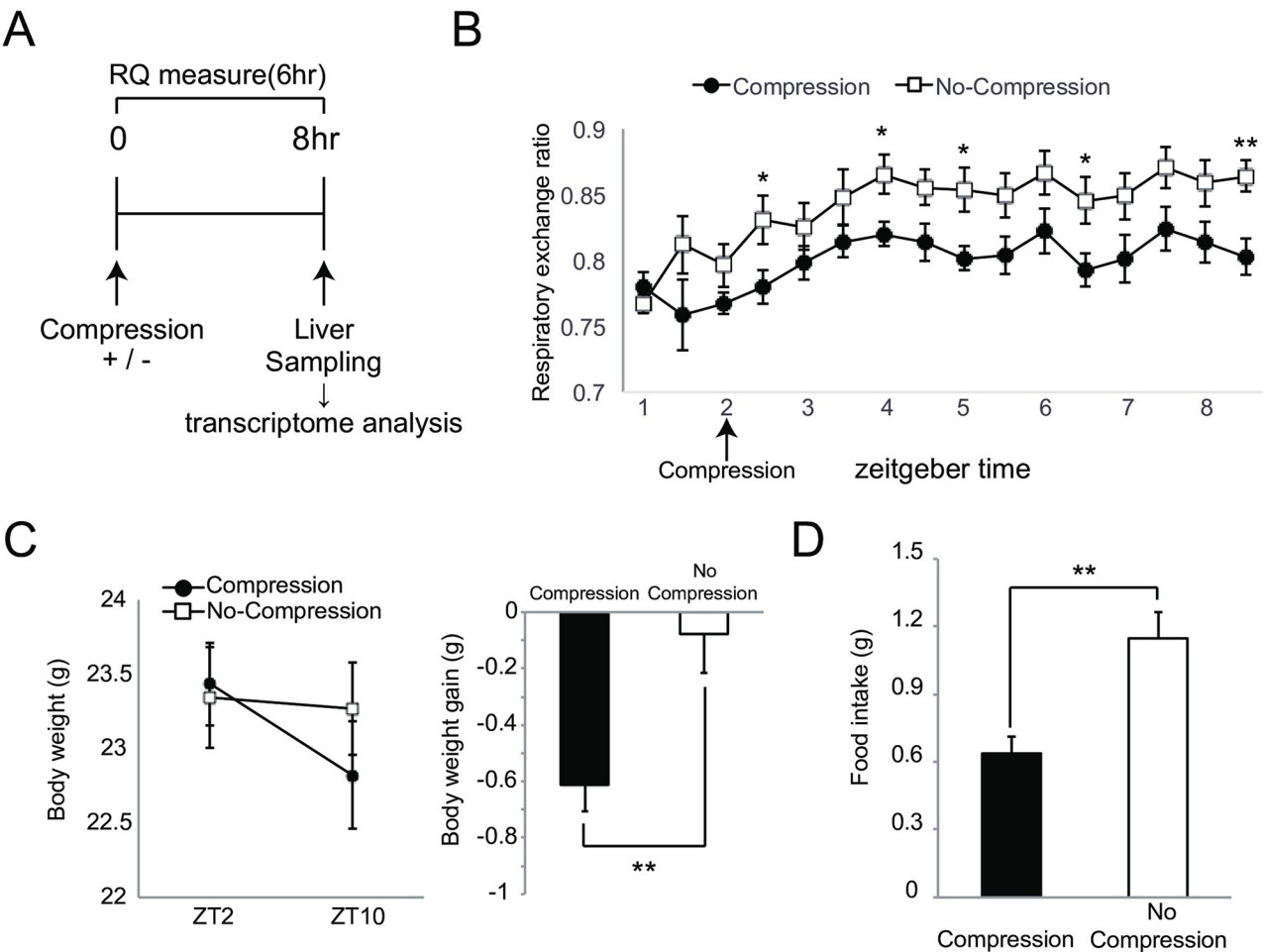

**Fig 3. Short-term effect (8hr on light period) of clothes compression load to the trunk.** A: Experimental protocol to investigate a short-term effect on clothes compression load (The acclimation condition before compression load is equal to Fig 1A protocol). B: Temporal change of respiratory exchange ratio (RER) after clothes compression load during 6hr. C: Measurement of body weight change by clothes compression load for 8hr (left panel is body weight change and right panel is body weight gain). D: Measurement of food intake amount by clothes compression load for 8hr. Each value is mean with SEM. Significant difference detected by unpaired t-test (*: p < 0.05, **: p < 0.01). n = 7 (no-compression group), n = 7 (compression group).

Family 27 Subfamily A Member 1 (Cyp27a1), ATP Binding Cassette Subfamily C Member 3 (Abcc3), and Bile Acid-CoA:Amino Acid N-Acyltransferase (Baat) were up-regulated in the compression group. In functional categories relating to immune response, DEGs relating to Chemokine (e.g. C-X-C Motif Chemokine Ligand 2 (Cxcl2), C-X-C Motif Chemokine Ligand 9 (Cxcl9)), interleukin cytokines (Interleukin 1 Receptor Antagonist (Il1rn), and Interleukin 33 (Il33)) were included; these gene expressions relating to immune response were down-regulated in the compression group (Fig 4B and S1 Table).

## 4. Discussion

In this study, we investigated changes in metabolic function via compression load to the trunk using mice as subjects. A previous study showed that Sprague–Dawley (SD) male rats attached with a body girdle over a long term experienced changes in body fat mass, body weight, and food consumption caused by serum leptin decreases mediated by mTOR signaling [13],

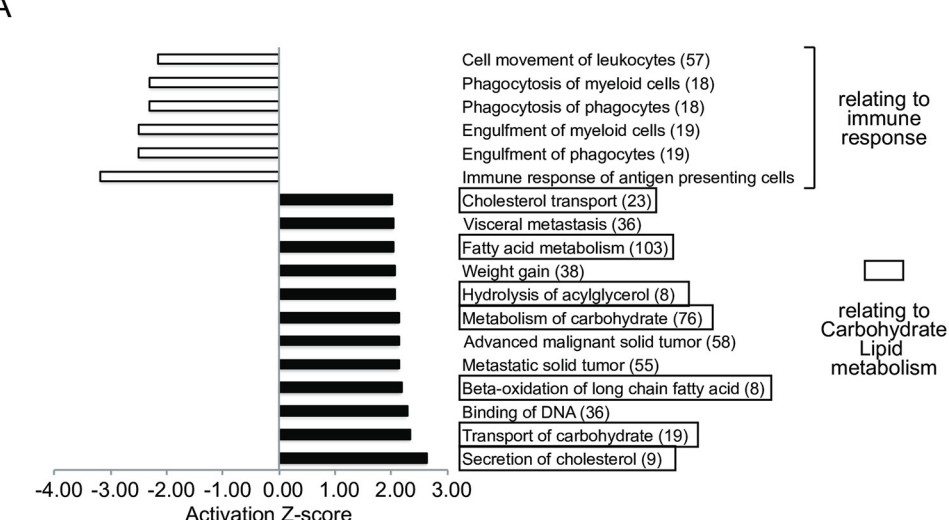

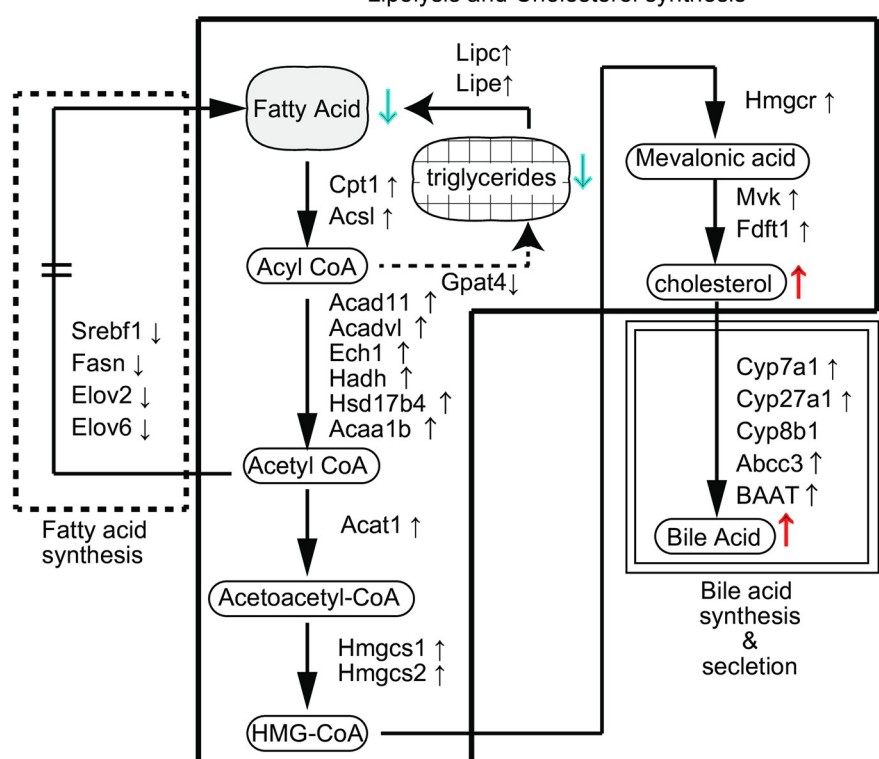

**Fig 4. Transcriptome analysis of liver changes caused by compression load to the trunk.** A: Top list of diseases and functions discovered by IPA core analysis in liver. B: A proposed mechanism in liver for clothes compression load to the trunk in light period is leading to lipolysis, cholesterol synthesis and bile acid secretory raise. Solid-line square means cholesterol synthesis process, dotted-line square means fatty acid synthesis process and double-solid-line square means bile acid synthesis / secretion process.

suggesting that the body girdle as a compression load to the trunk influences metabolic function. However, several studies reported that differences in degree of compression load value and compression period induced several different effects on cognitive function, motor function, and immune response [12, 14–17]. These reports suggest that the control of compression load value is an important factor in evaluating effects of compression load. To evaluate an effect on compression load to mouse, we made the optimal adjustable compression clothes for mice. We confirmed that adjustable compression clothes did not inhibit their common behavior patterns such as grooming and rearing. As a result of the application of a constant compression load to the trunks of mice, we observed that several effects were restricted to the light period (ZT2 to ZT11), which is an inactive phase for mice. Interestingly, some new effects were observed in the dark period (ZT12 to ZT23), the active phase for mice. Hence, the effects of compression load to the trunk indicated that these phenomena induced by compression load are closely linked to life rhythm because several metabolic changes observed were more prevalent in the inactive phase than in an active phase which demands more energy. To elucidate mechanisms of metabolic changes produced by the compression load to the trunk, we collected liver samples eight hours after compression load and performed transcriptome analysis. According to IPA analysis of disease and functions using DEGs in the liver, we found that several enriched functional categories were significantly affected in the compression group compared with the no-compression group. Those were related to metabolism function adjustment and were mainly suppression of fatty acid synthesis, activation of cholesterol synthesis process and activation of bile acid synthesis/secretion. Also, activation of lipolysis by compression load occurred because the gene expression of the hepatic lipase enzyme which hydrolyzes stored triglycerides to free fatty acids, Lipase E (Lipe) and Lipase C (Lipc), were increased in livers of the compression group compared with the no-compression group. Generally, internal cholesterol content is strictly controlled. Thus, it is likely that activation of lipolysis processes and cholesterol synthesis were induced by homeostatic maintenance of hepatic cholesterol concentrations caused by a food consumption decrease in an inactive phase. On the other hand, suppression of the functional categories including several DEGs relating to immune response were confirmed in the compression group. Those were decrease of the phagocytic activity and decrease of the migratory ability in immune cells. In general, it is known that the decrease of the phagocytic activity in immune cells is induced by stress [18]. It is also known that the corticosterone in response to stress has a function of lipolysis through activation of catabolic effects [19]. Therefore, it was suggested that the compression load to the trunk influenced metabolic function, e.g., lipolysis, as a kind of stressor.

Taken together, we revealed that compression loads applied to the trunk influenced lipid metabolism through lipolysis processes in an inactive phase. Recently, the effect of compression load has attracted a lot of attention in many fields such as treatment of lower extremity varicose veins, fatigue alleviation, and healthcare. Furthermore, when the effect of lipolysis through compression load is combined with choosing functional foods, further synergistic effects are expected because some functional foods including nobiletin also induce lipolysis activation [20]. Therefore, we will evaluate the combined effect of wearing compression clothes and consuming functional foods.

## Supporting information

**S1 Table. Disease and biofunctions analysis of the differentially expressed genes in the liver.**
(XLSX)

**S1 Fig. Ratio of tissue to body weight (g / g).** Left, liver: right, epididymal adipose tissue. Values are mean with SEM, n = 7 in each group.
(TIF)

## Author Contributions

**Conceptualization:** Kousuke Shimada, Masakatsu Nohara, Fumika Shinozaki, Midori Tatsuda, Takayuki Watanabe, Asuka Kamei, Keiko Abe.

**Data curation:** Kousuke Shimada, Masakatsu Nohara, Fumika Shinozaki, Asuka Kamei, Keiko Abe.

**Formal analysis:** Kousuke Shimada, Masakatsu Nohara, Fumika Shinozaki, Asuka Kamei, Keiko Abe.

**Funding acquisition:** Kousuke Shimada, Midori Tatsuda, Takayuki Watanabe, Asuka Kamei.

**Investigation:** Kousuke Shimada, Masakatsu Nohara, Fumika Shinozaki, Asuka Kamei, Keiko Abe.

**Methodology:** Kousuke Shimada, Masakatsu Nohara, Fumika Shinozaki, Midori Tatsuda, Takayuki Watanabe, Asuka Kamei, Keiko Abe.

**Project administration:** Kousuke Shimada, Masakatsu Nohara, Fumika Shinozaki, Asuka Kamei, Keiko Abe.

**Validation:** Kousuke Shimada.

**Visualization:** Kousuke Shimada, Masakatsu Nohara, Fumika Shinozaki, Asuka Kamei, Keiko Abe.

**Writing – original draft:** Kousuke Shimada, Masakatsu Nohara, Fumika Shinozaki, Asuka Kamei, Keiko Abe.

**Writing – review & editing:** Kousuke Shimada, Masakatsu Nohara, Fumika Shinozaki, Asuka Kamei, Keiko Abe.

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
