## [Decision Letter · Decision Letter 0]

28 Mar 2022

PONE-D-21-16430The effects of compression load to the trunk on lipid metabolism in an inactive phase.PLOS ONE

Dear Dr. Kamei,

Thank you for submitting your manuscript to PLOS ONE. After careful consideration, we feel that it has merit but does not fully meet PLOS ONE’s publication criteria as it currently stands. Therefore, we invite you to submit a revised version of the manuscript that addresses the points raised during the review process.

Specifically, reviewer 1 raised a couple of concerns including statistical analysis and animal research details. Please address all the comments.

We look forward to receiving your revised manuscript.

Kind regards,

Jianhong Zhou

Associate Editor

PLOS ONE

Journal Requirements:

3. Thank you for stating in your Funding Statement: "A part of this study was funded by DIANA co ltd. and this study was also supported by a research grant from JSPS KAKENHI Grant Numbers 15H05346 (to A. K.), 16K12734 (to K. S.), 18K05514 (to A. K.), 19K14047 (to K. S.)."

4. Thank you for stating the following in the Competing Interests section: "This study was funded by DIANA co ltd. T.W and M.T are employees of DIANA co ltd. The sponsor had no control over the interpretation, writing, or publication of this study."

We note that you received funding from a commercial source: [Name of Company]

Reviewers' comments:

Reviewer's Responses to Questions

**Comments to the Author**

1. Is the manuscript technically sound, and do the data support the conclusions?

Reviewer #1: Yes

Reviewer #2: Yes

2. Has the statistical analysis been performed appropriately and rigorously? 

Reviewer #1: No

Reviewer #2: I Don't Know

3. Have the authors made all data underlying the findings in their manuscript fully available?

Reviewer #1: Yes

Reviewer #2: Yes

4. Is the manuscript presented in an intelligible fashion and written in standard English?

Reviewer #1: Yes

Reviewer #2: Yes

5. Review Comments to the Author

Reviewer #1: This study about the compression loading on the murine trunk and related metabolism changes is interesting. This topic is worth to do more study to present more solid information for the understanding and evaluting of this aspect. And, the writing of this manuscript is standard and suitable for reading.

There're questions about the study shown below.

About the concept and the design.

Refer to the metabolism issues, it's well known that the processing is a various long term effects accumulated outcome. In this study, the compression loading lasted only for one day or eight hours. Are there any limitations restricted a long term loading on the animals? Even a 1 week continuous or intermittent treatment could be more convincing about a metabolic topic.

In addition, since it's a study about metabolism as highlighted in the title of this draft, it will be nice if any kind of metabolism related animal models could be utilized, more direct information could be collected for analyzing then.

About the term, animal ethics, and statistics

Please make sure the term concept is correct.

MSCs are not only stem cell of bone cells, and they are not real stem cells according to the definition of stem cell.

“Mice were anesthetized using 2.5% isoflurane at a flow rate of 1 L/min, and the liver samples were collected.” Since there's no euthenizing steps involved, do you mean the tissues collected while the animals are just been anesthetized, still alive?

How many samples used in the transcriptome analysis of liver? There's no N number shown.

Reviewer #2: The manuscript "The effects of compression load to the trunk on lipid metabolism in an inactive phase" investigated the effects of compression load to the trunk on mice metabolism. The findings were interesting, and provided useful information about compression load on health.

1. The authors should discuss why such a short time compression load have the remarkable effects on lipid metabolism in liver, and if a similar effects could be expected on a chronic compression load.

2. The animal numbers used in each experiments should be mentioned in the Method section.

6. PLOS authors have the option to publish the peer review history of their article (what does this mean?). If published, this will include your full peer review and any attached files.

Reviewer #1: No

Reviewer #2: No

---

## [Author Response · Author response to Decision Letter 0]

17 May 2022

Answer to Reviewer: 1, 

Thank you for your very valuable comments and useful suggestions to our manuscript entitled “The effects of compression load to the trunk on lipid metabolism in an inactive phase.” We revised the manuscript according to your comments. The major revisions are as follows:

Reviewer #1: This study about the compression loading on the murine trunk and related metabolism changes is interesting. This topic is worth to do more study to present more solid information for the understanding and evaluting of this aspect. And, the writing of this manuscript is standard and suitable for reading.

There're questions about the study shown below.

About the concept and the design.

Refer to the metabolism issues, it's well known that the processing is a various long term effects accumulated outcome. In this study, the compression loading lasted only for one day or eight hours. Are there any limitations restricted a long term loading on the animals? Even a 1 week continuous or intermittent treatment could be more convincing about a metabolic topic.

 In addition, since it's a study about metabolism as highlighted in the title of this draft, it will be nice if any kind of metabolism related animal models could be utilized, more direct information could be collected for analyzing then.

Authors' response:

 Thank you for your very valuable comments.

 In this study, we made adjustable compression clothes, but compression of the clothes had to be adjusted by manual. These clothes loosened over time because of losing body weight by compression, and had to be re-tightened under anesthesia to maintain the pressure. To avoid re-anesthesia repeatedly, since anesthesia has no small effect on metabolism, the experiments were conducted within a short time to maintain the clothes pressure.

 We think it is also important to validate using metabolism related mouse models as you suggested, and this is the next issue to be conducted.

 According to reviewer’s comment, we modified the text in the manuscript.

Page 6-7.

These clothes loosened over time because of losing body weight by compression, and had to be re-tightened under anesthesia to maintain the pressure. To avoid re-anesthesia repeatedly, since anesthesia has no small effect on metabolism, the experiments were conducted within a short time to maintain the clothes pressure.

Reviewer #1:

About the term, animal ethics, and statistics

Please make sure the term concept is correct.

MSCs are not only stem cell of bone cells, and they are not real stem cells according to the definition of stem cell.

Authors' response:

 We modified the text in the manuscript according to reviewer’s comment.

Page 4

For instance, the cells which constitute bones, including osteoblasts and osteoclasts have a mechanosensing network [1,2] and additional compression load to osteoblasts induce RANKL expression change and adjusts the bone morphological process [3–5].

Reviewer #1: “Mice were anesthetized using 2.5% isoflurane at a flow rate of 1 L/min, and the liver samples were collected.” Since there's no euthenizing steps involved, do you mean the tissues collected while the animals are just been anesthetized, still alive?

Authors' response:

 We modified the text in the manuscript according to reviewer’s comment.

Page 8

Mice were anesthetized using 2.5% isoflurane at a flow rate of 1 L/min, and after cervical dislocation, the liver samples were collected. These liver sections were immediately frozen after excision using liquid nitrogen.

Reviewer #1: How many samples used in the transcriptome analysis of liver? There's no N number shown.

Authors' response:

 We modified the description in the manuscript according to reviewer’s comment.

Page 6

The compression load value was measured using the pressure measuring system (AMI3037-SB; AMI-Techno, Tokyo, Japan) for the compression group (n = 20) and the no-compression group (n = 20) at the time of compression condition change.

Page 7

In the respiratory gases analysed for 24 hours, the compression group (n = 14) and the no-compression group (n = 12) were compared, and the analysis for six hours, n = 7 in the compression-loaded and the no-compression group, respectively, were compared.

Page 8

The samples used for DNA microarray were n = 7 in the compression-loaded and the no-compression group, respectively.

Figure/Table Legends

Fig. 1

B, n = 20 (no-compression group), n = 20 (compression group): C and D, n = 12 (no-compression group), n = 14 (compression group), respectively.

Fig. 3

n = 7 (no-compression group), n = 7 (compression group)

 

Answer to Reviewer: 2, 

Thank you for your very valuable comments and useful suggestions to our manuscript entitled “The effects of compression load to the trunk on lipid metabolism in an inactive phase.” We revised the manuscript according to your comments. The major revisions are as follows:

Reviewer #2: The manuscript "The effects of compression load to the trunk on lipid metabolism in an inactive phase" investigated the effects of compression load to the trunk on mice metabolism. The findings were interesting, and provided useful information about compression load on health.

1. The authors should discuss why such a short time compression load have the remarkable effects on lipid metabolism in liver, and if a similar effects could be expected on a chronic compression load.

Authors' response:

 Thank you for your very valuable comments.

 We discussed in the discussion paragraph on the mechanisms of that short time compression load has the remarkable effects on lipid metabolism in liver: the homeostatic responses according to the lowered food intake and the stress responses.

 It would be difficult to discuss chronic effects because it was not clear that these events were maintained for a long term. We think it is also important to validate chronic effects, and this is the next issue to be conducted.

Reviewer #2: 

2. The animal numbers used in each experiments should be mentioned in the Method section.

Authors' response:

 We modified the descriptions in the manuscript according to reviewer’s comment.

Page 6

The compression load value was measured using the pressure measuring system (AMI3037-SB; AMI-Techno, Tokyo, Japan) for the compression group (n = 20) and the no-compression group (n = 20) at the time of compression condition change.

Page 7

In the respiratory gases analysed for 24 hours, the compression group (n = 14) and the no-compression group (n = 12) were compared, and the analysis for six hours, n = 7 in the compression-loaded and the no-compression group, respectively, were compared.

Page 8

The samples used for DNA microarray were n = 7 in the compression-loaded and the no-compression group, respectively.

Figure/Table Legends

Fig. 1

B, n = 20 (no-compression group), n = 20 (compression group): C and D, n = 12 (no-compression group), n = 14 (compression group), respectively.

Fig. 3

n = 7 (no-compression group), n = 7 (compression group)

---

## [Editor Report · Decision Letter 1]

16 Jun 2022

The effects of compression load to the trunk on lipid metabolism in an inactive phase.

PONE-D-21-16430R1

Dear Dr. Kamei,

We’re pleased to inform you that your manuscript has been judged scientifically suitable for publication and will be formally accepted for publication once it meets all outstanding technical requirements.

Kind regards,

Hualin Wang

Guest Editor

PLOS ONE
---

## [Editor Report · Acceptance letter]

22 Jun 2022

PONE-D-21-16430R1 

The effects of compression load to the trunk on lipid metabolism in an inactive phase 

Dear Dr. Kamei:

I'm pleased to inform you that your manuscript has been deemed suitable for publication in PLOS ONE. Congratulations! Your manuscript is now with our production department. 

Kind regards, 

on behalf of

Dr. Hualin Wang 

Guest Editor

PLOS ONE